# Cytogenetic and Biochemical Responses of Wheat Seeds to Proton Irradiation at the Bragg Peak

**DOI:** 10.3390/plants12040842

**Published:** 2023-02-13

**Authors:** Lacramioara Oprica, Gabriela Vochita, Marius-Nicușor Grigore, Sergey Shvidkiy, Alexander Molokanov, Daniela Gherghel, Anda Les, Dorina Creanga

**Affiliations:** 1Biology Faculty, Alexandru Ioan Cuza University, 20A Carol I Bd., 700506 Iasi, Romania; 2Institute of Biological Research—Branch of NIRDBS, 47 Lascar Catargi Street, 700107 Iasi, Romania; 3Faculty of Medicine and Biological Sciences, Stefan cel Mare University of Suceava, 13 University Street, 720229 Suceava, Romania; 4Dzhelepov Laboratory, Joint Institute for Nuclear Research, 6 Joliot-Curie Street, 141980 Dubna, Russia; 5Physic Faculty, Alexandru Ioan Cuza University, 20A Carol I Bd., 700506 Iasi, Romania

**Keywords:** cytogenetics, proton irradiation, Bragg peak, abiotic stress

## Abstract

The present study aimed to evaluate the morphological, cytogenetic and biochemical changes in wheat seedlings as affected by seed exposure to a proton beam at the Bragg peak. The average energy of the proton beam was of 171 MeV at the entrance into the irradiator room while at the point of sample irradiation the beam energy was of 150 MeV, with the average value of the Linear Energy Transfer of 0.539 keV/μm and the dose rate of 0.55 Gy/min, the radiation doses being of the order of tens of Gy. Cytogenetic investigation has revealed the remarkable diminution of the mitotic index as linear dose-response curve as well as the spectacular linear increase of the aberration index. Analyzing some biometric parameters, it was found that neither dry matter nor water content of wheat seedlings was influenced by proton beam exposure. Studying the biochemical parameters related to the antioxidant defense system, we found that the irradiation caused the slight increasing tendency of peroxidase activity as well as the decreasing trend in the activity of superoxidedismutase in the seedlings grown from the irradiated seeds. The level of malonedialdehyde (MDA) and total polyphenols showed an increasing tendency in all seedling variants corresponding to irradiated seeds, compared to the control. We conclude that the irradiation clearly induced dose-response curves at the level of cytogenetic parameters together with relatively slight variation tendency of some biochemical parameters related to the antioxidant defense system while imperceptible changes could be noticed in the biometric parameters.

## 1. Introduction

Ionizing radiation contributed to the evolution of life on Earth 4 billion years ago [1]. These are electromagnetic or corpuscular radiations, with energy of quanta particles, which are capable of detaching an electron from any atom or molecule, within the radiation target.

The interaction of a proton beam with matter has several possible mechanisms—Coulombian interactions with electrons and nucleus of the reached atoms (resulting in loss of proton kinetic energy), inelastic nuclear reaction (when the incident proton enters into on track nucleus and another nuclear particle is emitted), as well as Bremsstrahlung radiation (when the proton suffers an energy loss, its trajectory being shifted and resultant photons being generated) [2,3]. The effect of ionizing radiation on biomolecules consists indirectly in the formation of reactive oxygen species (ROS), e.g., hydroxyl radical, hydrogen radical, superoxide, hydrogen peroxide, when interacting with the aqueous environment of the body, which could be further catalyzed by metal ions. In addition, reactive nitrogen species are generated when the radicals already formed combine with the nitric oxide and results in peroxynitrite anion, peroxynitrous acid, nitrogen dioxide and dinitrogen trioxide. These radicals can distort molecules that could also be altered directly when S–H, N–H, O–H, or C–H bonds are broken. Moreover, mitochondrial membrane potential can be injured following irradiation which could lead to problematic evacuation of mtDNA in the cytosol [4].

As mentioned by other authors [5], in biological applications, the proton energy in the range of 65–260 MeV results in LET (Linear Energy Transfer) values of about 0.4–1 keV. The relative biological effectiveness (RBE), defined as the ratio between the dose of the reference radiation and the dose of proton radiation to produce the same biological effect, for these therapeutic uses is usually placed in the domain 1.1–1.2 at the position of SOBP (Spread Out Bragg Peak). Some studies were conducted to assess the influence of proton energy on DNA structure that could lead to mutations, and indirectly to cancer or cell death. The most frequently occurring lesions were found to be either single-strand breaks or more critical double-strand breaks, which could create clustered lesions if the broken spots are located close to each other.

At the level of the nucleus of the cell, the most common induced types of defects (the starting point for later maladies) after exposure to ionizing radiation are expulsed micronuclei and rupture of nuclear envelope [6]. Studies conducted on proton irradiated human cells with a dose-rate of 0.5 Gy/min showed the specific defects induced on metaphases, specifically simple chromatid breaks, isochromatid breaks and dicentrics for which the determined aberration values were 3, 5 and 1, respectively, per 100 analyzed cells [7].

Currently, the intensity of ionizing radiation used in experimental research is much lower compared to that which penetrated the early Earth [8].

The interest in finding and understanding the effects of ionizing radiation in living organisms was intensified after a series of nuclear accidents (Chernobyl and Fukushima-Daiichi), with the field of radioecology developing and becoming able to detect and evaluate the effects of ionizing radiation in animals, plants and bacteria from the ecosystems, as well as the way to counteract them by plants or animals [9]. The effects that appear after the impact of radiation can be visible immediately, or sometimes the response can be after a latent period that is variable from a few minutes to a few decades depending on the irradiation dose and the intrinsic organism radiosensitivity. The different structure and metabolism between animal and plant cells make the latter more resistant to the action of radiation [10]. However, the radiation impact on mutations induction varies among taxa, with plants evidencing a higher effect than animals [11].

The exposure of seeds to ionizing radiation is used in mutation plant breeding programs to obtain new genotypes with improved qualitative and quantitative characters of many crop species. There are different types of ionizing radiation that have various energy levels and thus can penetrate cells to different degrees [12]. Gamma radiation is the physical mutagen used by plant breeders being proven that radiation has a role in increasing the genetic variability of the specie [13]. More than that, ionizing radiation through the production of free radicals may affect the germination, growth, and plant yield. In addition, high doses of gamma irradiation through stress signals have adverse effects on the physiological and biochemical traits of plants [14,15].

It is possible that protons had a role in the evolution of plants, being a main component of cosmic rays and also used for mutagenesis of plants. Moreover, the studies carried out on plants have focused especially on the effect of the ion beam; recently, the influence of proton beam irradiation has been used in mutagenesis [16,17]. The spectrum and/or frequency of mutations can be influenced by the dose used for irradiation, the type of plant tissue, and the period of time during which they act [18]. The link between the appropriate irradiation dose and mutation frequency is useful to minimize other harmful effects on plant survival and reproduction, although there is no analysis to support the relationship between irradiation dose and mutation frequency [19]. Physical strategies (such as ionizing radiation: X-rays, gamma rays, electron beams, proton beams, and heavy ion beams) used for seed priming are promising compared to traditional methods because they are time-saving, more efficient, more environmentally friendly, and there is greater certainty to improve yield in contrast to conventional methods (use of water or different chemicals) [20].

Proton particle radiation is a major part of solar particle events (about 87%) and consists of protons (hydrogen nuclei) with a certain energy (1~1000 MeV) which can stimulates organisms in a non-thermal manner [20]. Protons are usually absorbed by the plant and cause various changes in the physiological (stimulates the absorption of water by the seeds and photosynthetic pigments) and biochemical parameters (increase in the level of ROS, antioxidants) and the development of the plant’s defense systems to strengthen the production of culture [21,22,23,24]. In addition, proton radiation characteristic of the “Bragg peak” has a higher level of biological effect in organisms because the densest energy is deposited at the end of the path [5].

The comparative study by Lee et al. [25] regarding the effect of two proton beam beams (45 MeV and 100 MeV) and gamma radiation on Cymbidium hybrid indicated a significant oxidative stress, especially at the higher dose of proton beams compared to gamma radiation. The obtained results by Lyu Jae-Il et al. [26] which used proton beam irradiation on the germination of tobacco and rice seeds showed a low growth of the irradiated seedlings, but no significant differences in the morphological changes as a result of this treatment. They concluded that the proton beam can be used as a mutagen and the beam size and beam detection system must be established. The study carried out by Kim et al. [27] indicated that by increasing the dose of the proton beam used for irradiation, it was found that the survival rate of the Chinese yam plant decreased. Another study carried out on two varieties of rice indicated that proton beam irradiation determined changes in starch properties, especially at high dosage irradiation [28].

The growth and productivity of cereals and crop plants is affected by different abiotic stresses (e.g., drought, temperature extremes, salinity and acidity of soil, light intensity, submergence, and anaerobiosis) due to the current climatic conditions in the context of global warming [29,30]. Extended research in the last decades regarding the use of gamma rays to improve agroecosystems through the development of high-quality cultures/germoplasms have shown that low doses (50–100 Gy) promote tolerance to abiotic stress. Moreover, gamma radiation is used to obtain mutations in order to increase tolerance to abiotic stress and, of course, the creation of disease-resistant varieties [31].

Nutritional requirements worldwide, but especially in underdeveloped countries, are met by a series of cereals, but predominantly by wheat. Attempts are being made to create new wheat genotypes that are protected from environmental stress that is constantly changing. Thus, gamma ray irradiations were carried out which determined morphological, physiological, biochemical, genetic and cytological changes in the tissues depending on the doses used [32].

Our previous research [24] regarding the pretreatment of wheat seeds with proton beams was observed by changes in biochemical parameters (relatively slight decrease in chlorophyll *a* and *b*, decrease in antioxidant enzymes superoxide dismutase, catalase and peroxidase, as well as slight increases in malondialdehyde) in 7-day-old wheat seedlings.

In order to study the effect of the proton beam at the level of the genetic material, especially on the chromosomes of the biological material, the aim of this work was to evaluate the cytogenetic and biochemical responses of 7-day-old wheat seedlings grown from seeds irradiated with a proton beam at the Bragg peak.

## 2. Results 

### 2.1. Plant Growth Parameters

The morphology and plant growth measurements of 7-days-old wheat seedlings as a response to proton irradiation at the Bragg peak are shown in Figure 1, Figure 2 and Figure 3. Thus, in Figure 1 and Figure 2 we present the results of the dry mass and water content estimated as mentioned in the corresponding chapter above. It is visible that the young seedling growth was very similar in all samples, either irradiated or non-irradiated ones as long as no practical variation could be emphasized nor in the dry weight (Figure 1) nor in the water content (Figure 2).

Growth parameters presented in Figure 3A–C also appeared to be not significantly affected by the seed exposure to the proton beam at the Bragg peak for the radiation dose range applied in this experiment.

There was no coherent trend of variation in root or shoot length, suggesting no clear effect of proton impact of the growth of the seedlings. The seedling growth for seven days almost consumed seed resources (Figure 3B) so that the measurements and biochemical assays were carried out at that time. In Figure 3C, the morphology of representative seedlings from each sample is presented. A discussion on the corresponding contents of dry substances and water accumulation follows in further detail in the chapter below.

### 2.2. Cytogenetic Investigation Results

To assess any changes occurring in the cell cycle progression in the wheat roots apical meristem, the mitotic index (MI—the percentage of cells in mitosis), the percentage of cells in each mitosis phase (prophase, metaphase, anaphase and telophase), as well as clastogenic vs. aneugenic effects were determined. Thus, the modulation of the mitotic index and the frequency of each mitotic phase are plotted in Figure 4.

We could notice the general tendency of decreasing either for prophase, metaphase, anaphase and telophase, the most evident being the diminution of cells in prophase—from about 8.5% to 6% that appeared to be a monotonous negative variation to the radiation dose increase. Consequently, the division rate diminution was evidenced as presented by means of the mitotic index (the percentage of mitotic cells), in Figure 5.

The MI cytogenetic parameter diminished remarkably from about 13% in the control samples to less than 9% (i.e., with 30%) for the radiation dose of 89.6 Gy. A linear decreasing trend could be emphasized (MI = −0.7435 × Dose + 13.659 with R^2^ = 0.8157) with a certain increase for the dose of 56 Gy compared to neighbor doses of 44.8 and 67.2 Gy, that could be assigned to possible genetic uncontrollable variability in the corresponding seed bag.

The percentages of mitotic aberrations and that of micronuclei are represented in Figure 6. There was a considerable enhancement of the mitotic aberration parameter from around 0.8 in the control sample to almost 14% (meaning more than ten times) in the samples exposed to the higher radiation dose used in the frame of this experiment, i.e., 89.6 Gy. We mention that non-zeroed level of mitotic aberration percentage in the control sample could be explained by the effect of the environmental radiation background. The formation of micronuclei appeared to occur beginning with the exposure dose of 44.8 Gy, their percentage reaching about 5.6 % for the dose of 89.6 Gy. Linear variation in each cytogenetic parameter in Figure 6 is represented, with linear correlation coefficients, R^2^, over 0.99 and over 0.88, respectively.

Analysis of abnormal mitoses revealed different types of chromosomal changes affecting root tip cells in proton irradiated samples. Thus, bridges (single and multiple) were the most widespread type of aberrations, followed by multipolar ana-telophases (such as tri- and tetrapolar A-T). C-metaphases with chromosomes spread in the cytoplasm and complex aberrations (association between several aberrations) were also identified, as shown in Figure 7.

### 2.3. Photosynthesis Pigments

The chlorophyll *a* content is an essential factor associate to photosynthesis. According to the representation below (Figure 8) there was a remarkable increase of around 33% in the chlorophyll *a* level of 7-day-old wheat seedlings for the relatively high radiation doses, compared to the control, e.g., from 0.35 to 0.46 mg/g for the chlorophyll *a*, and from 0.05 to 0.09 mg/g for the chlorophyll *b*. In addition, it should be mentioned that an increase in the content of chlorophyll *a* and chlorophyll *b* was noticed, compared to the control and in the variants exposed to radiation of 67.2 Gy, by 52.3 and by 95.50%, respectively. At the same time, by contrast, the content of carotenes appeared considerable diminished (up to ten times) for the same samples corresponding to that relatively high proton dose (over 60 Gy), at doses 67.2 Gy and 89.6 Gy the decrease being of 88.32% and 89.92%, respectively, than control.

It is important to also comment the ratio of the chlorophyll contents (Figure 9) as this is taken as indicator of photosynthesis efficacy. This parameter was found to decrease clearly in the samples corresponding to the proton irradiation, with up to 25% for the highest radiation dose range used in this experiment (around 90 Gy) that suggested the impairing of the photosynthesis system PSII hosting enzyme systems involved in the photosynthesis complex process. One could associate these results with the hypothesis of radiation genotoxic effect in the irradiated seeds where possibly certain genes were affected and so the corresponding enzymes from cell chloroplasts.

Thus, there is no stimulation of the photosynthesis as least according to this chlorophyll related parameter, on the contrary there was a negative influence on the seedling development, even if the grown parameters discussed above apparently have not suggested it.

### 2.4. Antioxidant Enzymes

In this experiment, changes in the ROS scavenging enzymatic activities (superoxide dismutase—SOD, catalase—CAT, and peroxidase—POD), were evidenced in 7-days-old wheat seedlings as response to proton irradiation at the Bragg peak applied to wheat seeds (Figure 10A–C).

According to Figure 10A, SOD activity was diminished progressively to the increase in radiation dose with *p* ˂ 0.05 (Figure 10A), while non-significant variations could be detected in the CAT activity (Figure 10B) and only a slight, statistically insignificant, increasing trend (*p* > 0.05) could be shaped for POD activity (Figure 10C).

Catalase (CAT; EC 1.11.1.6) does not demand any reducing agent for its activity, being involved in a two-step reaction. If the concentration of H_2_O_2_ formed in presence of SOD is low, then this enzyme can remain in resting-state. In our study, the catalase activity was almost unchanged in the irradiated samples compared to the control one—kept at the natural background, the positive or negative variations remaining in the limits of the standard deviation (Figure 10B). It could be said that CAT activity was not adjusted accordingly to ROS level increase under radiation effect (it would be expected the enzyme biosynthesis to be stimulated) since probably proton irradiation has significantly affected the genes responsible for catalase synthesis.

Plant peroxidases (POD; EC 1.11.1.7) belong to Class III peroxidase enzymes and operate at the extracellular surface for scavenging H_2_O_2_. Both, POX and CAT, act simultaneity with SOD to prevent formation of harmful ROS by both •O_2_^−^ and H_2_O_2_ through a Haber-Weiss reaction [33]. Peroxidase activity seems to be slightly increased in the proton beam exposed samples (Figure 10C) from 3.5 to about 4.06 U/mg proteins, but this is in the limits of standard deviation (*p* > 0.05) and could not sustain statistically significant variations (Figure 10A).

### 2.5. MDA Content and Non-Enzymatic Antioxidants

MDA content (as a marker for lipid peroxidation intensity) in 7-day-old wheat seedling variants after exposure to a proton beam at the Bragg peak was clearly enhanced in the samples corresponding to seed irradiation (*p* < 0.05), from 36 to about 65 nmol/mg (about 75% positive variation), which indicates the more intense lipid peroxidation—apparently surprising, as long as the antioxidant enzyme activities were slightly influenced (Figure 11).

Aiming to determine the non-antioxidant responses of wheat seeds to proton beam irradiation (22.4–89.6 Gy), the polyphenol content in 7-day-old seedlings was measured (Figure 12). Polyphenols and flavonoids are non-enzimatic secondary metabolites that protect cells against the oxidative effects of ROS [34]. In the present study, polyphenol level was found to increase from 1.26 to around 2.08 mgGAE/g for proton dose around 60 Gy at the Bragg peak (*p* ˂ 0.05), this variation being possibly correlated with the enhancing of the MDA level presented above.

## 3. Discussion

### 3.1. Plant Growth Parameters

In other studies, the fresh weight of shoots and roots decreased with increasing doses of irradiation. Thus, Im et al. [35] found that, for higher proton doses than those used in our experiment, the dry weight of the shoots and roots of the varieties (Daepungkong Pungsannamulkong and Kwangankong) of soybean (*Glycine max* L. Merr.) was affected, being decreased, as corresponding to proton beam doses of 200 Gy and 300 Gy. Arena et al. [36] found that leaf dry weight of *Phaseolus vulgaris* L. after 22 days after exposure to X-rays was affected only for the dose of 100 Gy, whereas leaf relative water content was not influenced by either X-ray dose. On the other hand, in another study, the increase in the dose of gamma radiation has led to the significant diminution of dry weight of wheat stem and root [37]. Additionally, in contrast with our results where the water content was not influenced by irradiation doses, the increase in gamma radiation dose determined the decrease in the relative water content in wheat (*Triticum aestivum* L.) seedlings [38].

We mention that the radiosensitivity of wheat studied by Ahumada-Flores et al. [39] indicated that gamma ray irradiation at doses lower than 100 Gy causes positive morphological changes in its growth, while higher doses (300, 400, 500 and 600 Gy) negatively affect plant growth and survival due to deleterious effects that reduce plant genome stability, such as chromosomal damages. Im et al. [35] found that low-dose proton beam irradiation did not affect the height of soybean plants compared to high doses of 100–300 Gy.

The use of a proton beam (50 and 100 Gy) in the pretreatment of rice seeds was also reported to stimulate seedling growth, both height and root length of seedlings [28]. In addition, the same authors found that irradiation with proton beam at 50 and 100 Gy caused an intensification of the plant height in the rice cultivars, meanwhile the doses of 200–800 Gy causing a decrease in the plant height. Similar observations were conducted on rice seedlings exposed to relatively low doses (20–40 Gy) of radiation consisting in their growth-stimulating effect [16]. It seems that our results support the assertion that radiation bioeffects depend a lot on the radiation characteristics and the features of the biological material.

### 3.2. Cytogenetic Changes

Usually, the clastogenic action is reflected by the appearance of fragments, bridges and micronuclei, while the aneugenic effect does not include chromosome/DNA damages but determines eu- or aneuploidy by altering the mitotic spindle function.

Chromosomal bridges, resulting by interchromatid or subchromatid connections, represent the structural changes that may result from homologous or non-homologous chromosomes interexchanges and may lead to dicentric chromosome formation being a consequence of a poor activity of replication enzymes [40]. Micronuclei, indicators of genotoxicity and chromosomal instability exerted by different physical/chemical agents, include one or more entire chromosomes or chromosome fragments, which appear as a result of incorrect or unrepaired DNA breaks or non-disjunction of chromosomes during cell division.

Aneugenic changes include laggard or vagrant chromosomes (characterized by atypical migration to the mitotic spindle poles), and star-like polar anaphase. It is supposed that the formation of lagging chromosomes is due to the inhibition of tubulin polymerization. Moreover, sticky chromosomes and C-mitosis are the consequence of aneugenic transformation [41,42]. The alteration of normal function of some proteins that ensure the optimal organization of nuclear chromatin can increase its adhesiveness, leading to abnormal metaphases and anaphases, chromosomal bridges and, finally, the inhibition of cytokinesis with formation of binucleated cells [43]. Giant cells may be polyploid cells that have occurred through alteration of microtubule organization and cytokinesis progress [44,45]. C-mitosis is the colchicine-like metaphase that occurs as a result of blocking the progression of metaphase to anaphase by inactivating of the spindle formation and, consequently, the cell becomes polyploid [46,47].

### 3.3. Photosynthetic Pigments

Previous studies [24] regarding the single and combined effect of salinity and proton irradiation on the content of photosynthetic pigments (*chlorophyll a*, chlorophyll *b* and carotenoids) in barley seedlings indicated their damage in the sense of their decrease because of the applied treatments. In another recent research [23], in all samples corresponding to wheat seeds exposed to the proton beam, low levels (about 25%) of chlorophyll *a* content in wheat seedlings were detected.

The reduction in photosynthetic pigments may be due to the harmful effect of oxidative stress induced by the free radicals generated by ionizing radiation. Generally, in stress conditions, the photosynthesis is affected by ROS which determine a direct decrease in the chlorophyll *a* and total chlorophyll concentration [48]. Moreover, it was found that chlorophyll *a* is more exposed to ROS than chlorophyll *b*, and the ROS inducing direct deterioration of chlorophyll *a* and total chlorophyll level under stress circumstances [48,49].

Marcu et al. [50], reported biochemical differences of photosynthetic pigments (chlorophyll *a*, chlorophyll *b*, carotenoids) contents in maize plants exposed to gamma radiation evidencing an inversely proportional relationship to radiation dose. Another study reported that both total chlorophyll and total carotenoid content were significantly lower at 100 Gy than for lower doses and for control, non-irradiated leaves of *Phaseolus vulgaris* L. after 22 days of radiation exposure. On the other hand, leaves irradiated at 0.3 and 10 Gy exhibited the levels of total chlorophylls and carotenoids similar to the control [36]. The study carried out by Abou-Zeid and Abdel -Latif [38], indicated a significant decrease in the content of chlorophyll *a* and chlorophyll *b* in wheat seedlings as result of the application of different doses of gamma radiation to the seeds together with the significant increase in carotenoid content.

### 3.4. Antioxidant Defense Systems

Superoxide dismutase (SOD; EC 1.15.1.1) catalyzes the dismutation of superoxide anion (•O_2_^−^) to form H_2_O_2_ and O_2_. In plants, SOD is considered as one of the major enzymatic systems to scavenge stress-generated free radicals of superoxide •O_2_^−^. We found a slight but significant diminution trend (*p* < 0.05) in SOD activity for the samples exposed to the Bragg peak proton beam, while in another recent study [23], SOD activity was found remarkable decreased (*p* < 0.05) compared to the control in all proton exposed samples (by around 40%). The irradiator used in that experiment was identical to that utilized in the present study as well as the radiation doses, but the moderator filters were not applied so, at that time, we worked with relatively higher radiation intensity and probably this is why the SOD activity was influenced more.

Im et al. [35,51] published similar results, regarding the effect of the proton beam (57 MeV) on soybean seeds (*Glycine max* L. Merr.), the authors highlighted the decrease in SOD and POD activities along with the increase in the germination rate, the increase in the content of MDA and chlorophyll in plants.

Regarding antioxidant enzymes revealed by us in the 7-day-old wheat seedlings, the decreasing tendency of SOD activity together with the slightly increasing trend of POD activity is in contrast with the results reported by Abou-Zeid and Abdel-Latif [38], which highlighted a significant increase in SOD and POD activities in 21-day-old wheat leaves following seed exposure to gamma radiation, as well as the significant decrease in CAT activity. One could see that both the radiation type and the seedling age were different in comparison to our experiment.

Additionally, in a prior recent study [23], we found that all doses of radiation applied to wheat seeds inhibited CAT and POD in wheat seedlings, thus evidencing that the proton beam at the Bragg peak from the actual experiment influenced less these enzymes biosynthesis—CAT being practically unchanged, and POD being only slightly stimulated.

Plant growth, development and productivity can be affected by environmental stresses such as biotic (pathogens, parasites, grazing) and abiotic (drought, flooding, salinity, low-high temperatures, radiation, heavy metal toxicity etc.) factors. These kinds of stress often perturb the homeostasis and ion distribution in cells of plants and causing osmotic stress that led to an enhancement of reactive oxygen species (ROS). As the effect of the ionizing radiation absorption in the living cells, direct disruption of atomic structures can occur, producing chemical and biological changes [52]. In addition, low LET radiation has a high penetration capacity and it is mainly responsible for the generation of ROS by ionizing atoms.

It is already known that low LET radiation deposits energy in the biological target differently from high LET radiation [53]. Basically, in the first case, the transferred energy is spread in a larger volume than in the second one, mainly by means of free radicals generated along the radiation trajectory. Thus, biological effects caused by DNA molecule attacked by free radicals are generally higher for high LET radiation. DNA is most probably broken—with single or more frequently double strand breaks and repair mechanisms are less efficient than for low LET radiation impact because they need to face larger number of free radicals concentrated in certain molecular target volume.

This means, for the same absorbed dose, the genetic changes are expected to be higher for high LET radiation.

Proton radiation is characterized by lower LET compared to heavy ions, but higher than photon radiation.

However, the Spread Out Bragg Peak proton radiation, i.e., at the Bragg peak, it is supposed to have higher relative biological effectiveness than the proton beam at the entrance region, before filtration, and also higher than photon radiation at the same absorbed dose. Some experimental results [54] evidenced the most cytotoxic effects at the Bragg peak, along with extended cytotoxicity found also very close (millimeter order region) to the Bragg peak.

In the studies of Einor et al. [55], using different biological systems, the authors presented a meta-analysis demonstrating that ionizing radiation, even at low doses, generates ROS. Thus, the radiation produces abundant ROS via water radiolysis, some strong oxidants, and free radicals, which includes short-life species, e.g., superoxide anion radical (O_2_•^−^), hydroxyl radical (OH•), singlet oxygen (1O_2_), and long-life species, such as hydrogen peroxide (H_2_O_2_), etc. [56,57]. These radicals are toxic for a wide range of biological macromolecules such as DNA, lipids, and proteins [20,58]. Among these radiolysis products, the hydroxyl radical is the most dangerous for living tissues because of its high reactivity [59,60].

At low levels, ROS are important and essential since they maintain normal cellular functions by regulating the expression of specific genes, modulating ion channel activities, etc. [61]. In biological systems, when there is a serious imbalance between production of ROS together with reactive nitrogen species (RNS) and antioxidant defense, cellular damage occurs which lead to the appearing of oxidative stress [62,63]. Moreover, ionizing radiation produces a high level of ROS, which may cause oxidative stress [12,64]. The high chemical reactivity of ROS makes them very effective weapons against most biomolecules [65]. Plants have developed complex mechanisms to resist adverse environments, such as the ROS scavenged enzymatic antioxidant system. Both the enzymatic and non-enzymatic systems are efficacious in the defense of plants against oxidative stress. To avoid oxidative stress and the ROS scavenging too, plants activate several antioxidant enzymes, involved in either preventing the Haber-Weiss reaction or the Foyer–Halliwell–Asada pathway, which reduces the H_2_O_2_ and utilizes the reducing potential of NADPH. The antioxidant defense system includes enzymes such as superoxide dismutase (SOD), catalase (CAT), ascorbate peroxidase (APX), glutathione reductase (GR) and glutathione-S-transferase (GST) [66,67]. On the other hand, the non-enzymatic defense strategy is based on the action of various secondary metabolites, e.g., plant pigments, (such as anthocyanins, and carotenoids) as well as ascorbic acid, glutathione and phenolic compounds, which are able of free radical removal [33,34].

The higher capacity to remove ROS is usually correlated with a stronger antioxidant enzyme activity. At present, the knowledge concerning the role of the antioxidant systems in protecting plants under ionizing radiation stress is controversial and limited to very few plant species [36]. One could consider also other antioxidant systems being affected by radiation impact except those antioxidant enzymes (catalase, peroxidase or superoxide dismutase). It could be involved secondary metabolites, the flavones or polyphenols, i.e., the corresponding gene ability to control their synthesis in the seedlings developed from the irradiated seeds.

Free radicals can generally determine an increase in membrane permeability or a loss of its integrity which causes an increase in MDA level. As reported by Lee et al. [25], the use of 45 MeV proton beam radiation with higher-LET, determined a slightly increased MDA content in *Cymbidium hybrid* RB001 [(*C. sinensis* × *C. goeringii*) × *Cymbidium* spp.] while the 100 MeV proton beam with lower-LET led to the higher increase in this parameter. In our previous study [23] regarding the effect of proton beam irradiation of wheat seeds, we detected a slightly increasing tendency for ration doses lower that 60 Gy followed by the diminution trend for higher doses.

On the other hand, another previous study on 7-days-old seedlings *Hordeum vulgare* evidenced the decreased MDA content at both proton beam irradiation with 3 and 5 Gy, by a rate of 12.55% and 10.41%, compared to the control [24].

These results are different to previous ones [23] regarding the same dose range and biological material but without radiation peak smoothing—indeed in that experiment the polyphenol content being diminished while in the present experiment was slightly increased—but this could be just because the irradiation at the Bragg peak seems to induce lower effects as result of the radiation beam smoothing. On the other hand, results agree with those of Abou-Zeid and Abdel -Latif [49] that reported the significant increase in the total content of phenolics and flavonoids in wheat leaves as response to the application of different doses of gamma radiation, the increase being 4.5 and 4.9 times, respectively, compared to the control. Thus, the MDA content enhancement should be probably caused by the diminution of the levels of other protective systems, e.g., flavones for instance, that we are going to investigate in the future research activity.

Consequently, lipid peroxidation seems to be significantly higher in the samples corresponding to the proton beam exposure than in the control, non-irradiated ones, since the antioxidant enzymes do not actually protect against free radicals—the catalase being practically unchanged following the irradiation, while the POD and SOD appeared to have only slight opposite sense variations so that only the reduced antioxidant action of the polyphenolic compounds would be correlated with the intensified oxidation of cell membrane lipids evidenced in this experiment.

It is rather difficult to explain how the chromosomal changes revealed in the frame of cytogenetic investigation have affected the synthesis of the antioxidant enzymes and non-enzymatic systems. Certain chromosomal injuries could be compensated during seedling growth through the contribution of the repair systems—being known that wheat is one of the most radioresistant plant species. It is possible that the irradiation effects, given by the disruption of chromosome fragments, etc., have resulted in the stimulation of some cellular mechanisms (e.g., polyphenols and chlorophylls biosynthesis) and in the inhibition of others, e.g., carotenes and, eventually, SOD biosynthesis.

## 4. Materials and Methods

### 4.1. Experimental Design of Plant Material and Growing Conditions

Seeds of *Triticum aestivum* cv. Global used in the present experiment were obtained from Territorial Inspectorate for Seed and Propagating Material Quality Iasi. Intact seeds, which were identical in size and color, and free from wrinkles, were chosen for Proton Irradiation at the Bragg peak. The study of wheat seedling tolerance to irradiation was conducted under laboratory conditions, based on completely randomized design with three replications.

### 4.2. Proton Irradiation at the Bragg Peak

Irradiation of seed samples was carried out at the proton beam (Figure 13A) of the phasotron of Join Institute of Nuclear Research in Dubna, Russia, in 15 June 2020.

The average energy of the proton beam at the entrance into the irradiator cabin was 171 MeV. The measurements were carried out by a semiconductor silicon detector. On the way to the beam in the cabin there was constantly a retarder made of PMMA with a thickness of 40 mm of equivalent thickness of water. The irradiation of samples was carried out with an additional moderator with a thickness of 12 mm equivalent to the thickness of the water, while the beam energy at the point of irradiation of the samples is 150 MeV, the average LET, i.e., dE/dx = 0.539 keV/μm. The beam energy was reduced using PMMA moderators of various thicknesses. Proton beam dosimetry is based on the recommendations of the International Atomic Energy Agency (IAEA) [68]. Dosimetric calibration of the beam at each point of the deep dose distribution was carried out by the PTW UNIDOS-E clinical dosimeter with TM30013 ionization chamber.

For uniform irradiation of objects of relatively large thickness, the natural Bragg peak of the proton beam can be transformed to Spread Out Bragg Peak (SOBP). The conversion was carried out using a set of four ridge filters. The modified depth dose distributions of the radiation beam are shown in Figure 13B.

The mechanism of interaction of ionizing radiation with the biological objects depends on the type and the energy of the ionizing radiation. It can be described using several radiobiological quantities, of which the LET (Linear Energy Transfer) is one of the most important. In the case of charged particle beams, the value of the LET depends on the ion charge and ion energy. Because the mean energy decreases and energy spectrum widens when the particle beam passes the target volume, the LET and biological effectiveness of the primary beam changes. We measured the LET spectra of our proton beam at different depths using the Liulin-4C semiconductor energy deposits spectrometer [69].

In this experiment, the seed irradiation was carried out at the SOBP of the proton beam, the dose rate being of about 0.55 Gy/min. (Figure 13C).

### 4.3. Growth Conditions

Twenty wheat seeds (for each irradiated sample and control, a non-irradiated one) were placed in sterile Petri dishes on wet filter paper and kept in an Incucell room (22.0 ± 0.5 °C, in the dark), in Iasi university laboratory up to emergence of the shoots (48 h). To reach the seedling stage, the dishes were placed in controlled conditions of temperature (24.0 ± 0.5 °C), humidity (90%) and lightness (light/dark cycle: 14 h/10 h). Biochemical analyses were performed on 7-day-old wheat seedling.

### 4.4. Plant Growth Parameters

The radiation induced inhibition of vegetative growth was estimated by comparing several growth parameters between the control seedlings and the irradiated ones, after 7 days of grow: stem length (SL, cm), fresh weight (FW, g), dry weight (DW, g), and water content percentage (WC%). Total green tissue weight (FW) as well as the dry weight (DW) were measured for each variant. Dry weight was obtained by treating samples at 65 °C for 72 h. The water content percentage was calculated according to Cornelissen et al. [70] as WC% = [(FW − DW) x 100].

### 4.5. Cytogenetic Analysis

The seeds germination was conducted on filter paper moistened with distilled water. Ten primordial roots (10–15 mm) were collected and fixed in freshly prepared ethanol:acetic acid (3:1) solution for 24 h, at room temperature. The next day the roots were washed, placed in 70% ethyl alcohol, and stored at 4 °C. For staining, the root tips were softened in 37% HCl:distilled water solution (1:1) for 25 min, being then maintained in modified carbol-fuchsin dye [71] in a refrigerator. For each variant, five microscope slides were prepared by squash technique [72] using all three individual root tips from each germinated seed by placing on the slide in one drop of 45% acetic acid [73]. Fifty microscopic fields (ten per each slide) were analyzed using Nikon Eclipse 600 microscope (20× objective, Nikon, Kyoto, Japan) to assess the (i) mitotic index (MI), (ii) frequency of division phases and (iii) rate of chromosomal aberrations in ana-telophase. The relevant photos were taken with the Nikon Coolpix 950 digital camera (1600 × 1200 dpi, Nikon, Kyoto, Japan) and using the 100× oil immersion objective.

### 4.6. Photosynthetic Pigments Analysis

Chlorophyll *a*, chlorophyll *b*, and carotenespigments from 7-days-old wheat seedlings were extracted with 80% acetone according to the Lichtenthaler method [74]. Optical density of supernatant was determined (using Shimadzu Pharma Spec 1600 UV-visible spectrophotometer, Shimadzu Pharma, Kyoto, Japan) using light extinctions, E, at the wavelengths of 663 nm, 645 nm, and 470 nm. The pigment levels were expressed as mg/g of fresh weight according to the following equations:Chlorophyll *a* (Chl a) = (12.21 × E_663_ − 2.81 × E_645_)
Chlorophyll *b* (Chl b) = (20.13 × E_645_ − 5.03 × E_663_)
Carotenoids = [(1000 × E_470_ − 3.27 × Chl a − 104 × Chl b)/227]

### 4.7. Antioxidant Defense System

#### 4.7.1. Antioxidant Enzymes Activities

##### Preparation of Enzyme Extracts

To obtain the crude enzyme extract, aliquots of 7-day-old seedling green tissue were grounded in sodium phosphate buffer solution (pH = 7.5). Homogenates were centrifuged at 15,000 rpm/min at 4 °C for 15 min, the supernatants being used for the estimation of superoxide dismutase (SOD), catalase (CAT), and peroxidase (POD) activities. The specific activity of each antioxidant enzyme was expressed as unit per mg proteins (U/mg of protein).

SOD activity was assayed following Winterbourn’s method [75] using the ability to inhibit the reduction of nitro blue tetrazolium (NTB) of the superoxide radicals obtained by reoxidated of photochemically reduced riboflavin [24]. One unit of SOD activity is taken as the enzyme amount necessary to give 50% inhibition of the reduction of NBT, recorded at 560 nm.

POD activity was assayed spectrophotometrically using the method described by Ranieri et al. [76]. This assay is based on measurement of the color intensity generated by the oxidation of o-dianisidine 1% with H_2_O_2_ in the presence of peroxidase, at 460 nm wave length (Shimadzu Pharma Spec 1600 UV-VIS device, Shimadzu Pharma, Kyoto, Japan). Enzyme activity is expressed as peroxidase units—one unit corresponding to the quantity of enzyme able to decompose 1 micromole of H_2_O_2_ per minute.

CAT activity was measured according to Sinha’s procedure with minor adjustments [77]. This method is based on chromium acid determination obtained by reduction of potassium dichromate in acidic medium, in the presence of non-decomposed hydrogen peroxide, at 570 nm. One unit of CAT activity is expressed as the quantity of enzyme needed to reduce 1 μmol H_2_O_2_ per minute.

#### 4.7.2. MDA and Non-Enzymatic Antioxidant

Malondialdehyde (MDA), which is one of the final products of polyunsaturated fatty acids peroxidation in the cells, was quantified in enzymatic extracts according to the method described by Hodges et al. [78]. Thus, 1 ml of the enzymatic supernatant was mixed with 2 ml of 0.5% thiobarbituric acid (TBA) solution. The mixture was thermostated at 95 °C (60 min) and cooled at room temperature, being then centrifuged at 12,000 rpm (10 min) to remove the interfering substances. The absorbance of the TBA-MDA complex was measured at 532 nm using an UV–Vis spectrophotometer.

Total polyphenols content was quantified by applying Folin-Ciocalteu method with some modifications [79]. The absorbance of the methanol extract was evaluated at 765 nm after reacting with the Folin-Ciocalteu reagent. The total polyphenolic content was expressed as milligram equivalents of gallic acid per gram of dry weight (mg GAE g-1 DW) (R^2^ = 0.99). For each seedling variant, three measurements were carried out for the calculation of the average values and standard deviation.

### 4.8. The Soluble Protein Assay

The soluble protein content was assessed with Bradford’s method [80] using bovine serum albumin as standard. This method involves the binding of Coomassie Brilliant Blue G-250 dye to the aromatic amino acid radicals, the light absorbance being estimated at 595 nm wave length (Shimadzu Pharma Spec 1600 UV-VIS device, Shimadzu Pharma, Kyoto, Japan). The results were expressed as mg protein per g fresh weight.

### 4.9. Statistical Analysis

All determinations were performed in three repetitions and the results were expressed as the mean values ± standard deviation. The statistical significance of the differences between irradiated and control variant was assessed by means of the Student t test. The *p*-values less than 0.05 were considered significant.

## 5. Conclusions

Irradiation with a proton beam at the Bragg peak has induced some undoubtable cytogenetic changes as demonstrated by the diminution of the mitotic index with about 30% as well as the increase in the mitotic aberration percentage by more than ten times. Chromosomal aberrations suggested the double strand breaks with the inactivation of some genes possibly responsible for the changes noticed further in some biochemical parameters, such as photosynthesis pigment levels and the antioxidant defense system. At the level of the morpho-physiological parameters, we found no statistically significant changes in the contents of dry matter and water nor in the lengths of seedling roots and shoots, as a response to the proton irradiation. Chlorophyll contents were found increased for the relatively high radiation doses used in this experiment (over 60 Gy) while the photosynthesis apparent efficacy showed diminution tendency (with more than 25%).

Certain variation trends were revealed in the activities of the antioxidant enzymes and the level of polyphenols. Membrane lipid peroxidation was evidenced for all radiation doses applied in present study by means of the MDA content that was enhanced about twice. Cereal seedling response to proton beam could be of interest for long duration space travels when plant cultivation could be needed out of space while cosmic radiation could never be totally controlled—even if it could be smoothed by space station protective walls.

## Figures and Tables

**Figure 1 plants-12-00842-f001:**
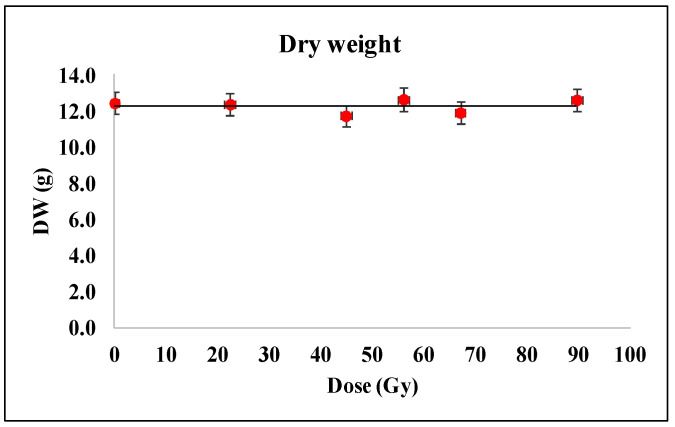
Dry weight (DW) in the wheat seedlings grown from the proton beam irradiated seeds. Twenty seedlings for each experimental variant were measured.

**Figure 2 plants-12-00842-f002:**
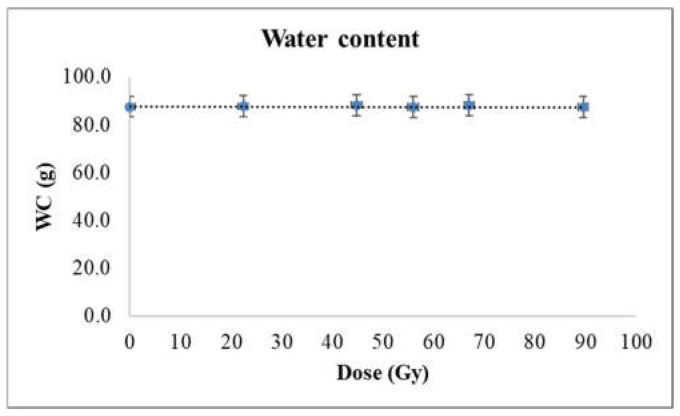
Water content (WC) in wheat seedlings grown from the proton beam irradiated seeds. Twenty seedlings for each experimental variant were measured.

**Figure 3 plants-12-00842-f003:**
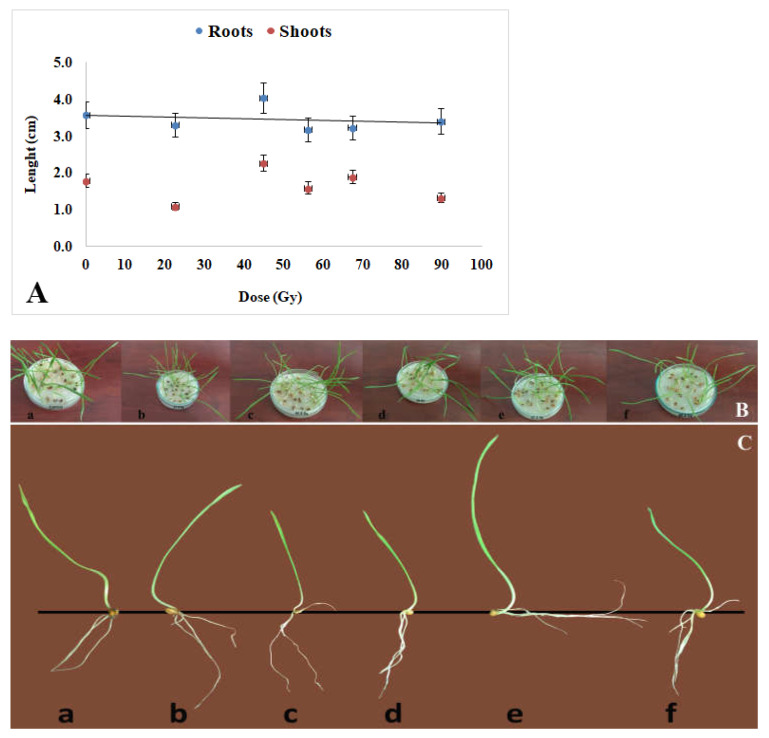
(**A**). Roots and shoots in the seedlings grown from the seeds exposed to a proton beam at the Bragg peak showed only slight variations, in the limits of standard deviation, with no distinct correlation with the applied radiation dose. Twenty seedlings for each experimental variant were measured. (**B**). General aspect of wheat seedlings (7 days old) grown in Petri dishes after proton treatment (a-Control; b-22.4 Gy; c-44.8 Gy; d-56 Gy; e-67.2 GY, f-89.6 Gy). (**C**). Wheat seedling morphology in each sample after 7 days of germination (a-Control; b-22.4 Gy; c-44.8 Gy; d-56 Gy; e-67.2 GY, f-89.6 Gy). Each dish hosted 25 seedlings and for each experimental variant, three repetitions were conducted.

**Figure 4 plants-12-00842-f004:**
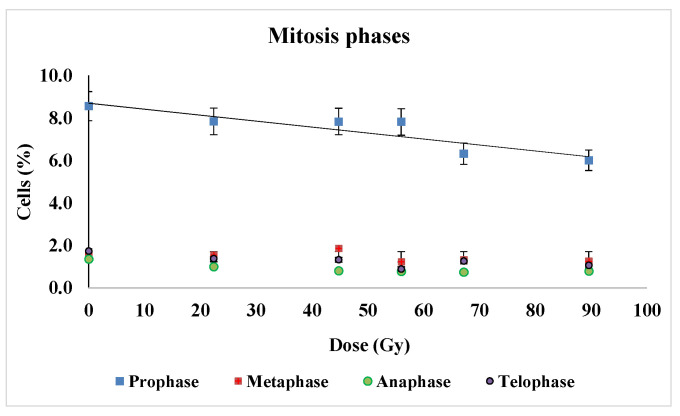
Cell percentage in mitosis phases in the proton beam irradiated samples at the Bragg peak. Average values and standard deviations were calculated for each experimental variant from five microscope slides that were analyzed, between 2300 and 3100 cells being found in each slide.

**Figure 5 plants-12-00842-f005:**
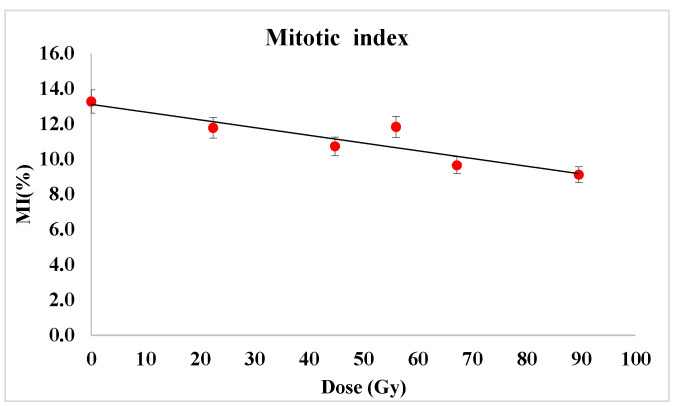
Mitotic index diminution following proton beam exposure. Linear dose-response was evidenced. From 2300 to 3100 analyzed cells from each of five microscope slides corresponding to each variant provided the data used for average value and standard deviation calculation.

**Figure 6 plants-12-00842-f006:**
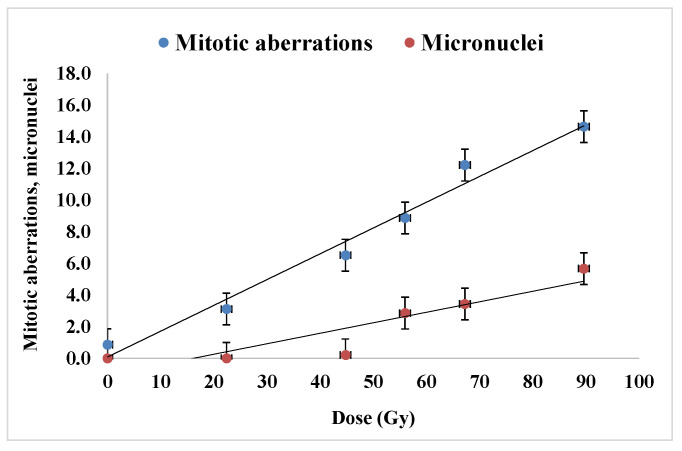
Linear increase of mitotic aberrations and micronuclei in the samples corresponding to proton beam exposure at the Bragg peak. For each experimental variant, between 2300 and 3100 data collected from the microscopy investigation were used.

**Figure 7 plants-12-00842-f007:**
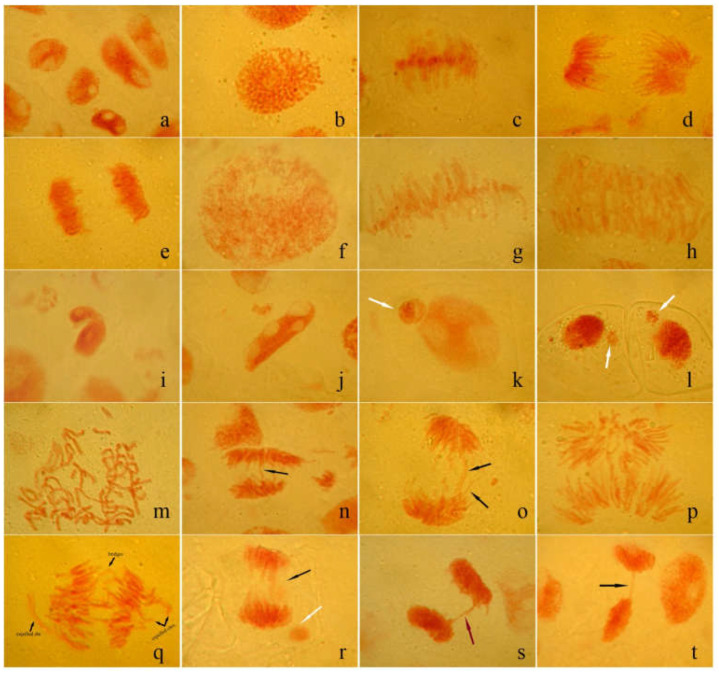
Normal and aberrant mitosis in *Triticum aestivum* L. root meristem cells after SOBP treatment. (**a**–**e**) normal inter-, pro-, meta-, ana- and telophase; (**f**–**h**) giant pro-, meta- and anaphases; (**i**) comma shaped interphase; (**j**) torpedo shaped interphase; (**k**,**l**) micronuclei with different sizes (white arrows); (**m**) C-metaphase; (**n**) A-T with single bridge (black arrow); (**o**) A-T with two bridges (each black arrow); (**p**) disaster shaped anaphase and multiple bridges; (**q**) complex aberration (A-T with bridges and expelled chromosomes); (**r**) late anaphase with bridges (black arrow) and micronucleus (white arrow); (**s**) overlap chromatids (red arrow); (**t**) telophase with single bridge (black arrow). From each of the five microscope slides corresponding to each experimental variant, between 2300 and 3100 cells were found and used for the cytogenetic analysis.

**Figure 8 plants-12-00842-f008:**
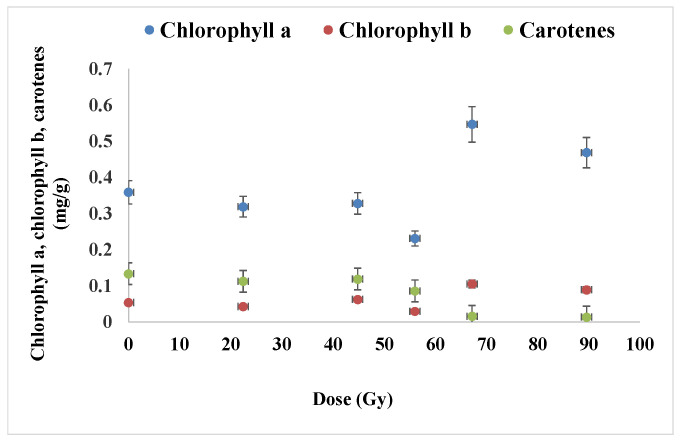
The influence of proton beam at the Bragg peak on the photosynthesis pigments. Three repetitions of the photosynthesis pigment assay were carried out for each experimental variant.

**Figure 9 plants-12-00842-f009:**
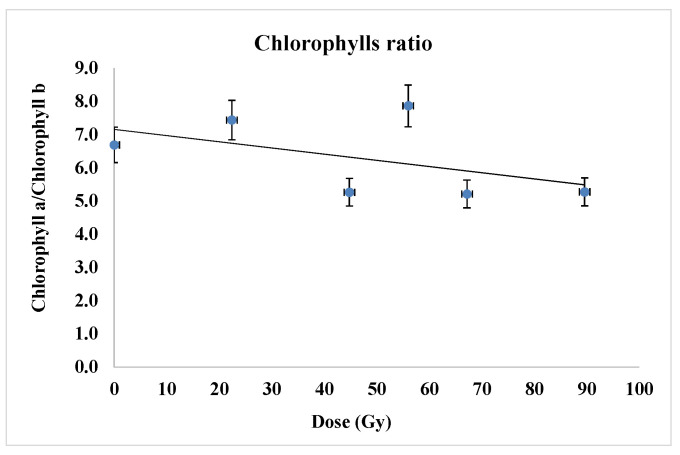
The influence of proton beam at the Bragg peak on the ratio of chlorophylls. Three assay repetitions were conducted for each experimental variant.

**Figure 10 plants-12-00842-f010:**
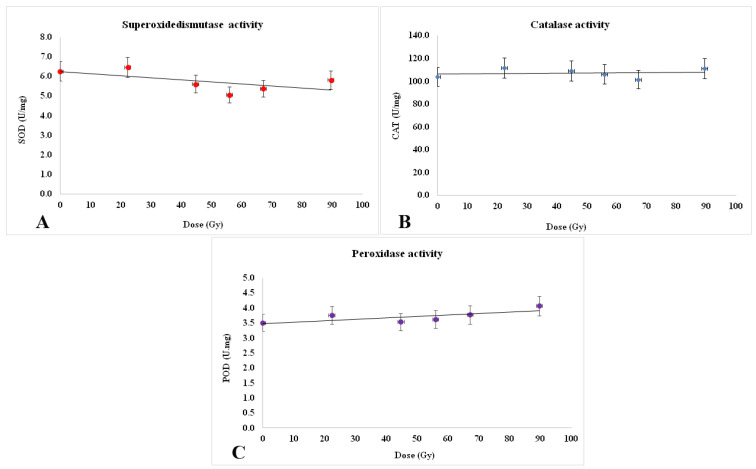
The activity of antioxidant enzymes in the samples corresponding to the proton irradiation at the Bragg peak. (**A**). Superoxidedismutase activity showed a slight but statistically significant decreasing tendency. (**B**). Catalase activity was found at approximately the same level in all samples. (**C**). Peroxidase was found to have a slight, statistically insignificant increase in the limits of standard deviation. Three repetitions of each enzyme assay were carried out for each experimental variant.

**Figure 11 plants-12-00842-f011:**
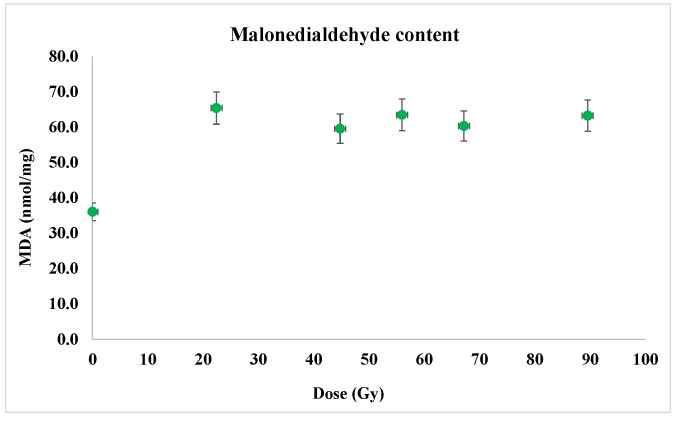
The content of malondialdehyde in the samples corresponding to the proton irradiation at the Bragg peak. Considerable increase in MDA in the samples corresponding to irradiated seeds was emphasized. For each experimental variant, three repetitions of the MDA content assay were carried out.

**Figure 12 plants-12-00842-f012:**
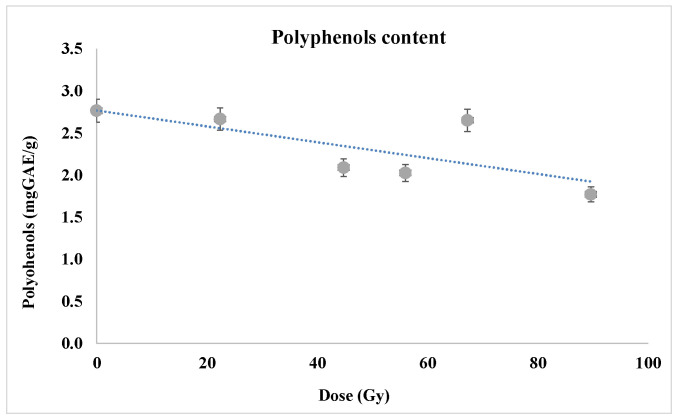
Polyphenol content in the samples corresponding to the proton irradiation at the Bragg peak. Linear dose-response was found with positive variations of the polyphenol content. For each experimental variant, three repetitions of polyphenol content assay were carried out.

**Figure 13 plants-12-00842-f013:**
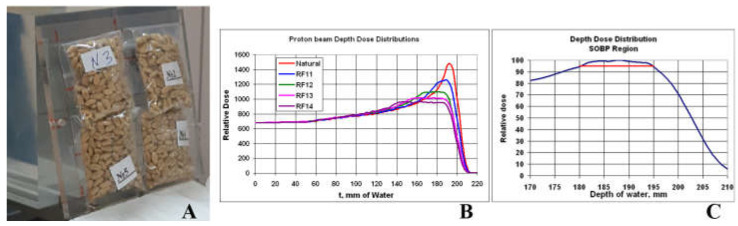
(**A**)**.** Wheat seed samples arranged for the irradiation. (**B**). Set of Spread Out Bragg Peaks (t- absorption thickness). (**C**). Region of the sample irradiation—depth of 180 mm of water.

## Data Availability

Data sharing not applicable.

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
