# Peer review of "Cytogenetic and Biochemical Responses of Wheat Seeds to Proton Irradiation at the Bragg Peak"

_plants, 2023, doi:10.3390/plants12040842_

Round 1
Reviewer 1 Report
The manuscript entitled "Cytogenetic and biochemical responses of wheat seeds to proton irradiation at the Bragg peak" presents biological effects of wheat in response to proton beam.
- Major commments
1. Seven-days-old seeding treated with various dose of proton beam displayed no growth inhibition, but mitotic index of samples was reduced with high dose treatment. Early seeding of plant can show no morphorogical changes after various irradiation. Thus, longer period (over 30 days) of observation after treatment should be required.
2. All Figure legend should include number of samples (plants, cells), experimental repeat, and/or biological repeat. Every figure legend should have a detailed explanation.
3. These are no SD (Standard Deviation) in Figure 6, 7, 9, 12
4. For figure 9, is this average number (y-axis) ?
5. Authors insist that peroxidase activity (Figure 15) was slightly induced by the proton beam irradiation. But all values are within SD. So, it is very hard to say induction of peroxidase activity among samples after irradiation.
6. Accurate description of Figure 13 should be required. Authors refered to another recent study, but scientific explanation of Figure 13 was not found.
7. It is necessary to clarify implications of this manuscriptin in the conclusion section.
8. The data was organized into one graph per Figure. It would be nice to show the data grouped.
- Minor comment
Please use subscript for H2O2, O2
Author Response
The authors of this paper thank to reviewers for their observation. Below the responses are given. Thank you!
The manuscript entitled "Cytogenetic and biochemical responses of wheat seeds to proton irradiation at the Bragg peak" presents biological effects of wheat in response to proton beam.
- Major commments
- Seven-days-old seeding treated with various dose of proton beam displayed no growth inhibition, but mitotic index of samples was reduced with high dose treatment. Early seeding of plant can show no morphological changes after various irradiation. Thus, longer period (over 30 days) of observation after treatment should be required.
Mitotic index was assessed at couple of days after germination while the seedling growth is presented at seven days when morphological changes among the seedlings appeared to be indistinguishable – but different situation could be expected later, in older seedlings. However, we did not succeed in keeping all the seedlings alive for longer time since they consumed almost all seed content. To observe seedling growth for longer time new nourishing substrate would be necessary, eventually in natural environment where, possibly uncontrollable environmental gradients could induce random changes that could overlap onto the irradiation effects.
- All Figure legend should include number of samples (plants, cells), experimental repeat, and/or biological repeat. Every figure legend should have a detailed explanation.
The required changes were done for each figure. Several figures were grouped as required below to other observation.
Figure caption
Figure 1 (former 4). Dry weight (DW) in the wheat seedlings grown from the proton beam irradiated seeds. Twenty seedlings for each experimental variant were measured.
Figure 2 (former 5). Water content (WC) in wheat seedlings grown from the proton beam irradiated seeds. Twenty seedlings for each experimental variant were measured.
Figure 3 (former 6). A. Roots and shots in the seedlings grown from the seeds exposed to proton beam at Brag peak showed only slight variations, in the limits of standard deviation, with no distinct correlation with the applied radiation dose. Twenty seedlings for each experimental variant were measured
B. General aspect of wheat seedlings (7 days old) grown in Petri dishes after proton treatment (a-Control; b-22.4 Gy; c-44.8 Gy; d-56 Gy; e-67.2 GY, f-89.6 Gy). C. Wheat seedlings morphology in each sample after 7 days of germination (a-Control; b-22.4 Gy; c-44.8 Gy; d-56 Gy; e-67.2 GY, f-89.6 Gy). Each dish hosted 25 seedlings and for each experimental variant three repetitions were done.
Figure 4. (former 7) Cell percentage in mitosis phases in the proton beam irradiated samples at Bragg peak. Average values and standard deviations were calculated for each experimental variant from five microscope slides that were analyzed, between 2300 and 3100 cells being found in each slide.
Figure 5 (former 8). Mitotic index diminution following proton beam exposure. Linear dose-response was evidenced. 2300 to 3100 analyzed cells from each of five microscope slides corresponding to each variant provided the data used for average value and standard deviation calculation.
Figure 6 (former 9). Linear increase of mitotic aberrations and micronuclei in the samples corresponding to proton beam exposure at Bragg peak. For each experimental variant between 2300 and 3100 cells were found and used.
Figure 7. (former 10) Normal and aberrant mitosis in Triticum aestivum L. root meristem cells after SOBP treatment. (a-e) normal inter-, pro-, meta-, ana- and telophase; (f-h) giant pro-, meta- and anaphases; (i) comma shaped interphase; (j) torpedo shaped interphase; (k-l) micronuclei with different sizes (white arrows); (m) C-metaphase; (n) A-T with single bridge (black arrow); (o) A-T with two bridges; (p) disaster shaped anaphase and multiple bridges; (q) complex aberration (A-T with bridges and expelled chromosomes); (r) late anaphase with bridges and micronucleus; (s) overlap chromatids (red arrow); (t) telophase with single bridge. From each of the five microscope slides corresponding to each experimental variant between 2300 and 3100 cells were found and used for the cytogenetic analysis.
Figure 8 (former 11). The influence of proton beam at Bragg peak on the photosynthesis pigments. Three repetitions of the photosynthesis pigment assay were carried out for each experimental variant.
Figure 9 (former 12). The influence of proton beam at Bragg peak on the ratio of chlorophylls. Tree assay repetitions were done for each experimental variant.
Figure 10 (former 13, 14 and 15). The activity of antioxidant enzymes in the samples corresponding to the proton irradiation at the Brag peak. A. Superoxide dismutase activity showed slight but statistically significant decreasing tendency. B. Catalase activity was found at approximately the same level in all samples. C. Peroxidase was found to have slight, statistically non-significant increase in the limits of standard deviation. Three repetitions of each enzyme assay were carried out for each experimental variant.
Figure 11 (former 16). The content of malondialdehyde in the samples corresponding to the proton irradiation at the Brag peak. Considerable increase of MDA in the samples corresponding to irradiated seeds was emphasized. For each experimental variant three repetitions of the MDA content assay were carried out.
Figure 12 (former 17). Polyphenol content in the samples corresponding to the proton irradiation at the Brag peak. and we supplemented with Linear dose-response was found with positive variations of the polyphenol content. For each experimental variant three repetitions of polyphenol content assay were carried out.
Figure 13 (former 1, 2 and 3). A. Wheat seed samples arranged for the irradiation. B. Set of Spread Out Bragg Peaks (t- absorption thickness). C. Region of the sample irradiation – depth of 180 mm of water.
- These are no SD (Standard Deviation) in Figure 6, 7, 9, 12
The required changes were done.
- For figure 9, is this average number (y-axis) ?
Yes, according to new figure caption.
- Authors insist that peroxidase activity (Figure 15) was slightly induced by the proton beam irradiation. But all values are within SD. So, it is very hard to say induction of peroxidase activity among samples after irradiation.
In Results section we wrote (with figure number updated)
According to Figure 10 A, SOD activity was diminished progressively to the increase of radiation dose with p˂0.05, (Fig. 10 A), while non-significant variations could be detected in the CAT activity (Fig. 5 B) and only slight, statistically non-significant, increasing trend (p>0.05) could be shaped for POD activity (Fig. 10 C).
In Results section we wrote
Peroxidase activity (Fig. 10 C) seems to be slightly increased in the proton beam exposed samples as from 3.50 to about 4.06 U/mg proteins, but this is in the limits of standard deviation (p> 0.05) and could not sustain significant statistically variations.
- Accurate description of Figure 13 should be required. Authors refered to another recent study, but scientific explanation of Figure 13 was not found.
The required change was done to the caption of the graph with the enzyme activities (now Figure 10 A, B, C) and, in the Results section, we wrote
Also, at Discussion section we insert: We found slight but significant linear diminution trend (p˂0.05) in SOD activity for the samples exposed to Bragg peak proton beam while in other recent study [23] SOD activity was found remarkably decreased (p˂0.05) compared to the control, in all proton exposed samples (with around 40%). The irradiator used in that experiment was identical to that utilized in the present study as well as the radiation doses, but the moderator filters were not applied so, at that time, we worked with relatively higher radiation intensity and probably this is why the SOD activity was influenced more.
- It is necessary to clarify implications of this manuscript in the conclusion section.
We wrote to Conclusion section
Seedling response to proton beam could be of interest for long duration space travels when plant cultivation could be needed out of space while cosmic radiation could never be totally controlled – even it could be smoothed by space station protective walls.
- The data was organized into one graph per Figure. It would be nice to show the data grouped.
- Minor comment
Please use subscript for H2O2, O2
The required changes were done all over the text.
Reviewer 2 Report
Title: Cytogenetic and biochemical responses of wheat seeds to proton irradiation at the Bragg peak
Compared to other radiation sources such as gamma rays or heavy ion-beams, there are not many studies on the biological effects or mutations when plants are irradiated with proton beams. This paper is intended to look at the morphological, cytological and biochemical changes in seedlings when wheat seeds are irradiated with proton beams aligned with the Bragg peak, and it is meaningful in terms of providing these basic data. However, while reading this article, I have a few questions.
1. Why, however, barley in the previous study of these authors was treated with a low dose of 0 to 5 Gy, although it was said that it was treated in parallel with NaCl treatment, not for the purpose of mutagenesis. But in this study, it is not clear whether relatively high doses such as 0, 22.4; 44.8, 56, 67.2 and 89.6 Gy were treated compared to at that time. A higher dose may be required for investigations to induce mutations, but there does not seem to be more test on this.
2. In order to select an appropriate dose for mutagenesis after radiation treatment or to see biological effects, most plants are generally investigated two weeks after irradiation, and germination rate and survival rate must be checked. I wonder why this study did not checked this.
3. In cancer treatment using ion beams, irradiation is performed in accordance with the Bragg Peak area in order to accurately irradiate only the cancer tissue cell, but I think when irradiating seeds that need to penetrate the seed, it may be better to use energy in the front area rather than the Bragg Peak area.
4. The LET value of this proton irradiation is 0.539 Kev/um, higher than that of gamma rays, but much lower than that of heavy ion beam irradiation. It seems that more discussion is needed in terms of differences in biological effects or mutagenesis among these radiation sources.
5. Looking at previous papers and this paper, I wonder if the purpose of the authors' research is to develop seed priming treatment by proton beam irradiation in addition to the discovery of basic knowledge.
Author Response
The authors of this paper thank to reviewers for their observation. Below the responses are given. Thank you!
Compared to other radiation sources such as gamma rays or heavy ion-beams, there are not many studies on the biological effects or mutations when plants are irradiated with proton beams. This paper is intended to look at the morphological, cytological and biochemical changes in seedlings when wheat seeds are irradiated with proton beams aligned with the Bragg peak, and it is meaningful in terms of providing these basic data. However, while reading this article, I have a few questions.
- Why, however, barley in the previous study of these authors was treated with a low dose of 0 to 5 Gy, although it was said that it was treated in parallel with NaCl treatment, not for the purpose of mutagenesis. But in this study, it is not clear whether relatively high doses such as 0, 22.4; 44.8, 56, 67.2 and 89.6 Gy were treated compared to at that time. A higher dose may be required for investigations to induce mutations, but there does not seem to be more test on this.
We renounced to that comparison (former ref. 24) as not useful.
- In order to select an appropriate dose for mutagenesis after radiation treatment or to see biological effects, most plants are generally investigated two weeks after irradiation, and germination rate and survival rate must be checked. I wonder why this study did not checked this.
The seedlings grown on watered paper support consumed almost all seed content and could not be kept alive for longer time – this is why we carried out biochemical assays at seven days (while cytogenetic investigation was performed on couple of day old seedlings when root meristems have first appeared).
- In cancer treatment using ion beams, irradiation is performed in accordance with the Bragg Peak area in order to accurately irradiate only the cancer tissue cell, but I think when irradiating seeds that need to penetrate the seed, it may be better to use energy in the front area rather than the Bragg Peak area.
We accomplished such an experimental study (without filter the radiation beam), the results being published last year in Romanian Journal of Physics, but now we searched to see, comparatively, the seed response to Bragg Peak irradiation.
- The LET value of this proton irradiation is 0.539 Kev/um, higher than that of gamma rays, but much lower than that of heavy ion beam irradiation. It seems that more discussion is needed in terms of differences in biological effects or mutagenesis among these radiation sources.
We wrote, in the Discussion section
It is already known that low LET radiation deposits energy in the biological target differently from high LET radiation [53] Basically, in the first case, the transferred energy is spread in larger volume than in the second one, mainly by means of free radicals generated along the radiation trajectory. Thus, biological effects caused by DNA molecule attacked by free radicals are generally higher for high LET radiation. DNA is most probably broken – with single or more frequently double strand breaks and repair mechanisms are less efficient than for low LET radiation impact because they need to face larger number of free radicals concentrated in certain molecular target volume. This means, for the same absorbed dose, the genetic changes are expected to be higher for high LET radiation. Proton radiation is characterized by lower LET compared to heavy ions, but higher than photon radiation. However, the Spread Out Bragg Peak proton radiation – i.e., at the Bragg peak, is supposed to have higher relative biological effectiveness than the proton beam at the entrance region, before filtration, and also higher than photon radiation at the same absorbed dose. Some experimental results [54] evidenced the most cytotoxic effects at the Bragg peak, along with extended cytotoxicity found also very close (millimeter order region) to Bragg peak.
- Looking at previous papers and this paper, I wonder if the purpose of the authors' research is to develop seed priming treatment by proton beam irradiation in addition to the discovery of basic knowledge.
The main purpose was to get basic knowledge on the seed response to proton beam irradiation.
Round 2
Reviewer 1 Report
Dear Authors
I think that the manucript is well improved after the first revision. I recommend publication of this manuscript.